# Current Pharmacological Treatment Options for Central Serous Chorioretinopathy: A Review

**DOI:** 10.3390/ph13100264

**Published:** 2020-09-23

**Authors:** Massimo Nicolò, Lorenzo Ferro Desideri, Aldo Vagge, Carlo Enrico Traverso

**Affiliations:** 1IRCCS Ospedale Policlinico San Martino, University Eye Clinic of Genoa, 16132 Genoa, Italy; lorenzoferrodes@gmail.com (L.F.D.); aldo.vagge@gmail.com (A.V.); mc8620@mclink.it (C.E.T.); 2Department of Neurosciences, Rehabilitation, Ophthalmology, Genetics, Maternal and Child Health (DiNOGMI), University of Genoa, 16148 Genoa, Italy; 3Macula Onlus Foundation, 16132 Genoa, Italy

**Keywords:** central serous chorioretinopathy, photodynamic therapy, OCT, OCTA, choroid

## Abstract

Central serous chorioretinopathy (CSC) is a common cause of visual impairment in patients generally aged 20 to 60 and it is characterized by acute or chronic neurosensory detachments of the retina. Although CSC resolves spontaneously in most cases, in some patients it may cause permanent visual impairment in the working population; for this reason, several approaches, including photodynamic therapy (PDT), subthreshold micropulse laser treatment and oral mineralocorticoid receptor antagonists, have been studied as first-line treatment options for CSC. To date, half-dose PDT has provided the most encouraging results in this regard, supported by large, multicenter, randomized clinical trials such as the “Prospective Randomized Controlled Treatment Trial for Chronic Central Serous Chorioretinopathy” (PLACE) trial; however, the role of novel possible non-invasive treatment options is attracting interest. This review article aims to discuss the current pharmacological treatment options investigated for the management of CSC, including aspirin, ketoconazole, beta blockers, rifampicin and many others. In particular, further evidence about oral mineralocorticoid receptor antagonists, firstly seen as promising non-invasive alternatives for treating CSC, will be provided and discussed in light of the recent “Eplerenone for chronic central serous chorioretinopathy in patients with active, previously untreated disease for more than 4 months” (VICI) trial results, which have largely resized their role as possible first-line oral treatment options for treating CSC.

## 1. Introduction

Central serous chorioretinopathy (CSC) is a common chorioretinal disorder characterized by an idiopathic retinal serous detachment of the retina, associated with one or multiple areas of leakage originating from the choroid through a defect in the retinal pigment epithelium (RPE), the outer blood–retina barrier [1]. CSC accounts for the fourth most frequent cause of retinopathy, after age-related macular degeneration, diabetic retinopathy and retinal vein occlusions [2]. The reported incidence of the disease is approximately 9.9 cases per 100,000 men and 1.7 per 100,000 women, showing an increased prevalence in men (almost six times higher) as compared with women [3]. Several risk factors have been associated with the pathogenesis of CSC; in particular, psychosocial stressors, a type A personality, glucocorticosteroid use, trait anxiety, endogenous hypercortisolism and pregnancy have been shown to display important roles in this regard [2,4,5].

Clinically, CSC presents with a central scotoma, which may be often associated with distorted vision (metamorphopsia) and alterations in color perception (dyschromatopsia); at the first visit, the best-corrected visual acuity (BCVA) generally ranges from 20/20 to 20/200 [6].

Many authors subdivide CSC into two clinical subtypes: the acute form (aCSC), which typically resolves within 3–4 months without need for being treated, and the chronic form (cCSC), which is characterized by the presence of persistent serous detachment visible by optical coherence tomography (OCT) for longer than 4–6 months; in some cases, cCSC may lead to permanent structural damage of the RPE and the photoreceptor cell layer, causing irreversible long-term visual impairment [7,8,9].

Although the diagnosis and follow-up of CSC is typically allowed by OCT examination, the role of multimodal imaging, including fundus autofluorescence (FAF), fluorescein angiography (FA), indocyanine green angiography (ICGA) and more recently OCT angiography (OCTA), is mandatory in order to better guide the management of the disease. Multimodal imaging in CSC is able to detect RPE decompensation and atrophy, choroidal hyperpermeability and possible choroidal neovascularization. In particular, the novel OCTA technique allows one to better characterize CSC pathogenesis, by detecting the low flow in the choriocapillaris, which may be due to compression by enlarged choroidal vessels typical of CSC [10,11].

Several treatment options, including photodynamic therapy (PDT) with verteporfin, subthreshold retinal laser treatment, transpupillary thermotherapy (TTT) and pharmacological therapy (in particular, oral mineralocorticoid-receptor antagonists) have been investigated for treating CSC; in this regard, some randomized clinical trials have shown that PDT and subthreshold retinal laser treatment are the most promising treatment options currently available. In this regard, Nicolò et al. showed in a multicenter study that the full reabsorption of subretinal fluid (SRF) was achieved in 83.9% and 100% of the eyes treated with half-fluence and half-dose PDT, showing an overall safe and effective profile for this procedure however, there is growing interest in the role of pharmacological therapy given its more conservative and non-invasive approach [12].

In this review we will discuss the clinical efficacies and safety profiles of the most promising drugs investigated for the management of CSC, and nonetheless, we will analyze the role of PDT, which can be considered a pharmacological treatment option.

## 2. Mineralocorticoid and Glucocorticoid-Receptor Oral Antagonists

The rationale of the use of mineralocorticoid (MR) and glucocorticoid (GR)-receptor antagonists relies on the increased levels of cortisol and the dysregulation in the endogenous MR levels reported in patients with CSC [13]. These increased cortisol levels reflect altered hypothalamic-pituitary-adrenocortical (HPA) axis activity, as often observed in individuals with symptoms of depression, anxiety and stress [14,15,16]; however, to date, no studies have investigated the role of antidepressant drugs in treating CSC.

In vivo studies on rats have revealed increased expression of MRs after corticosteroid administration [10]. Likewise, corticosterone, a glucocorticoid agent, has been shown in rat models to increase choroidal thickness, which represents a typical feature of patients with CSC [17].

Given this background, MR antagonists spironolactone and eplerenone have been employed in pilot studies for treating patients with CSC with encouraging results [18]; however, some limitations are present with this class of drugs: close monitoring of renal function and potassium levels should be performed before starting therapy with MR antagonists, considering the risks such as cardiac arrhythmia associated with hyperkalemia. Thus, MR antagonists should be prescribed to patients with potassium serum levels <5.5 mEq/L and a creatinine clearance rate of >30 mL/m [19].

### 2.1. Eplerenone

Eplerenone is a specific MR antagonist, which is composed of a 9,11-epoxide group and was first licensed for heart failure management; due to its molecular structure, eplerenone use is associated with a reduced incidence of hormone-associated spironolactone adverse events [20,21].

In a non-randomized pilot study, Bousquet et al. treated 13 patients with CSC with oral eplerenone at the dosage of 25 mg/day for a week followed by 50 mg/day for 1 or 3 months. They found significant decreases in central macular thickness (CMT) after 1 and 3 months (*p* < 0.05 and *p* < 0.01, respectively), a significant reduction of SRF after 3 months (*p* < 0.01) and a significant improvement from baseline in best-corrected visual acuity (BCVA) (*p* < 0.001) [18].

In another retrospective study on 24 patients with cCSC resistant to conventional therapy, a treatment regimen with oral eplerenone (25 mg/day) for 1 week, followed by a protracted daily dose of 50 mg, was adopted. It was revealed that 29% of the patients experienced a complete resolution of SRF after an average of 106 days of treatment. Moreover, OCT images showed that the presence of a baseline integrity of the ellipsoid zone and RPE were related to better visual outcomes [22,23].

By contrast, another prospective, double-blinded, placebo-controlled study randomized 17 patients with persistent SRF due to CSC to either eplerenone 50 mg/day or placebo for 3 months, with a 3-month follow-up period. There was no significant difference in terms of BCVA and SRF between the two groups, showing that eplerenone was not superior to placebo [24].

Interestingly, in another prospective study, it was shown that the absence of choroidal neovascularizations (CNV) at OCTA examination and the presence of a hotspot at ICGA were positively associated with a complete response to oral eplerenone treatment (*p* < 0.001 and *p* = 0.002, respectively) [25].

Recently, there was a large, multicenter, randomized, double-blinded, parallel-group, placebo-controlled VICI trial that randomized 114 patients with cCSC to either eplerenone (n = 57) or placebo (n = 57). After 12 months, the mean BCVA was 79.5 letters in the placebo group and 80.4 letters in the eplerenone group, revealing a mean difference of 1.73 letters (95% CI −1.12 to 4.57; *p* = 0.24). Results from this large-scale, randomized trial suggested that eplerenone clinical efficacy in patients with cCSC was not superior to placebo [26].

### 2.2. Spironolactone

Spironolactone is a potassium-sparing diuretic acting as a binding competitor of aldosterone and approved for treating congestive heart failure and primary hyperaldosteronism [27].

Several studies have reported the potential beneficial effects of spironolactone in improving BCVA, decreasing CMT and resolving SRF in patients with CSC [28].

In a prospective, randomized, placebo-controlled crossover study, Bousquet et al. randomized 16 patients to either spironolactone 50 mg or placebo once a day for 30 days, followed by a 1-week wash-out period and then a cross-over to either placebo or spironolactone for another 30 days. A significant reduction in SRF in spironolactone-treated eyes was found compared with placebo (*p* = 0.04); however, no differences were found in BCVA between the two groups [28].

Another prospective study enrolled 21 patients with cCSC and treated them with 25 mg spironolactone twice a day; it was found that 15 eyes (71%) had decreased SRF on OCT 12 months after the start of the treatment [29].

A prospective, randomized study on 30 eyes with aCSC, revealed that 56% (n = 10/18) patients treated with spironolactone experienced a complete SRF resolution at two months in comparison with 8% (n = 1/12) in the control group (*p* < 0.05) [30].

In another prospective, placebo-controlled clinical trial on 60 patients with CSC, spironolactone was found to be slightly superior to eplerenone in improving BCVA, whereas both drugs had similar clinical outcomes in promoting the reabsorption of SRF [31].

Further, randomized clinical trials should better describe spironolactone clinical efficacy for treating CSC.

### 2.3. Mifepristone

Mifepristone (also called “RU-486”) is a GR and progesterone receptor antagonist with high-affinity, commonly adopted in gynecological clinical practice for inducing pharmaceutical abortion [32].

Nielsen et al. performed an uncontrolled study on 16 patients with cCSC treated with mifepristone at 200 mg daily for 12 weeks. The authors found that 44% of the patients (n = 7/16) improved by five or more letters in BCVA, and the same percentage of them had improved OCT findings. Moreover, no serious adverse events (AEs) were reported [33].

Recently, a randomized, double-masked placebo-controlled study (STOMP CSC) on 30 patients with cCSC was presented at ARVO 2018. Patients were randomized to mifepristone 300 mg daily, mifepristone 900 mg daily or placebo for 4 weeks. After the 4-week treatment period, they were observed for other 4 weeks without treatment. They reported a significant reduction in CMT of 82 μm (*p* < 0.05) and an improvement in BCVA of 3.6 letters (*p* < 0.05) in both mifepristone groups, as opposed to the non-significant reductions in CMT (47 μm, *p* = 0.45) and BCVA (0.7 Letters, *p* = 0.64) in the placebo group. Between the two treatment groups, no differences were found [34].

In spite of the encouraging clinical outcomes extrapolated from this trial, further evidence should be provided, in order to describe mifepristone clinical efficacy.

## 3. Other Oral Pharmacological Treatment Options for Treating CSC

In the last few years several oral pharmacological treatment options (including antioxidants, beta-blockers, carbonic anhydrase inhibitors and melatonin) have been investigated for the management of CSC; however, most of the evidence has been provided by small, uncontrolled clinical studies [7]. Nonetheless, considering that most of the cases of aCSC resolve spontaneously, these results should be taken with caution.

### 3.1. Antioxidants

The role of inflammation and oxidative stress has been thought to be involved in the pathogenesis of CSC; given this rationale, antioxidant agents have been studied for the management of this disease [8].

A randomized, placebo-controlled study treated patients with aCSC with either high-dose antioxidant tablets (study group A) or placebo tablets (control group B) for 3 months or until the complete resolution of subretinal fluid; however, after 3 months, no significant differences were reported between the two groups in both BCVA and CMT [35].

In another study, a curcumin-phospholipid (lecithin) oral formulation was given to patients with CSC for 12 months. It was found that 100% showed no decrease in BCVA, and 61% of them experienced a significant improvement (*p* = 0.0001); however, the lack of a control group and the absence of a CSC clinical description (acute or chronic) represent important limitations of this study [36]. Further comparative studies should provide evidence in favor of the efficacy of antioxidants for CSC treatment.

### 3.2. Aspirin

In patients with CSC, higher levels of plasminogen activator inhibitor have been reported in comparison with healthy subjects, and thus, it has been thought that hypercoagulability may represent a possible pathogenic factor related to CSC onset [37].

In a prospective study, Caccavale et al. treated 109 patients with CSC with daily aspirin, 100 mg for 1 month, followed by 100 mg on alternate days for 5 months. They showed that the aspirin-treated group achieved better visual outcomes for the first and third months from the onset of CSC in comparison with controls; however, an important limitation of the study is represented by a historical group used as controls [38]. Hence, the evidence of a possible role of aspirin for treating CSC is still limited.

### 3.3. Beta-Blockers

Beta-blockers are commonly used for systemic hypertension and anxiety disorders [39]. The already mentioned association between stress, type-A personality and CSC onset has suggested that beta-adrenergic agents may represent as a possible therapeutic option for CSC [40].

In a prospective, double-blinded study on 48 patients with aCSC, the examined group was treated with metipranolol 10 mg twice per day and it was compared with placebo (n = 25/48). They revealed no significant differences in duration of CSC between the two groups (*p* = 0.341) [41].

Likewise, in another study, Browning et al. found no clinical efficacy of nadolol for treating CSC [42]. By contrast, in a case report of two patients with cCSC treated with metoprolol (a selective β1 receptor blocker), there was complete resolution of SRF; however, the nature of the study with only two patients does not allow one to draw any conclusion about the efficacy of metoprolol for CSC treatment [43]. Overall, there is no evidence of a possible role of beta-blockers for the treatment of CSC.

### 3.4. Carbonic Anhydrase Inhibitors

Acetazolamide is carbonic anhydrase inhibitor commonly adopted for treating glaucoma and idiopathic intracranial hypertension. The rationale of acetazolamide adoption for treating CSC would reside in its inhibition of carbonic anhydrase IV (?) in the retinal pigmented epithelium (RPE), which could cause the resorption of subretinal edema and restore the physiological polarization of RPE cells [44].

In a retrospective study, 15 patients were treated with oral acetazolamide and showed a time reduction until complete resolution of SRF than the control group (3 weeks versus 8 weeks, respectively); however, no differences were found in BCVA and the rate of recurrence between the two groups [45].

Similarly, another retrospective study divided 45 patients with aCSC into an acetazolamide group (group 1, n = 20/45) and control group (group 2, n = 25/45). They showed no significant improvement in BCVA in both the groups (*p* = 0.083 and 0.183). Moreover, SRF height and choroidal vascular decreased significantly after 3 months in both groups (all *p* < 0.05). Thus, although acetazolamide displayed no significant effect on functional or anatomical status in patients with CSC, it shortened the time for SRF absorption [46].

In this regard, larger-scale and well-designed clinical studies should better define acetazolamide role for treating CSC.

### 3.5. Finasteride

Androgens, and in particular testosterone, have been postulated to play a role in the pathogenesis of CSC [47]. Finasteride is a 5-alpha-reductase, which works by inhibiting dihydrotestosterone, and it is commonly adopted for the treatment of benign prostatic hypertrophy and androgenic alopecia [48]. In a prospective pilot study on five patients with CSC, the finasteride-treated group showed no significant differences in terms of BCVA gain and CMT reduction after 6 months (with a 3-month treatment period). Moreover, patients had full resolution, and after treatment discontinuation, an increase in SRF was reported in four patients (n = 4/5) [47].

Differently, a retrospective study on 23 patients with cCSC treated with finasteride showed that 76% of them had complete SRF resolution after a 15-month follow-up period [49]; however, due to the possible onset of relatively common side effects (such as the loss of libido) and the poor evidence provided, finasteride should not currently be considered a viable treatment option for CSC.

### 3.6. Melatonin

Melatonin is involved in the physiological regulation of circadian rhythm and it has been postulated to exert a possible positive protective effect on CSC [50].

In a prospective case series, 13 patients with cCSC were treated for 1 month: eight patients received 3 mg melatonin orally and five were given placebo. They reported an improvement in BCVA in 87.5% of the patients in the melatonin group (7 of 8 patients, *p* < 0.05) and all patients showed a reduction (*p* < 0.01) of CMT [51]. No other studies have been performed investigating the role of melatonin for CSC treatment yet; thus, further studies are needed to prove its potential clinical efficacy.

### 3.7. Helicobacter pylori Eradication

Currently, there is increasing evidence about the potential pathogenic role of *Helicobacter pylori* (*H. pylori*) infection on CSC onset [52]. *H. pylori*, a gram-negative bacterium, is associated with peptic ulcers, gastric cancer and MALT lymphoma development, and it is commonly treated with antibiotics, including metronidazole, amoxicillin and/or clarithromycin [53]. In CSC, it is thought that some *H. pylori*-related antigens may cause an autoimmune response to homologous host proteins in the endothelium of the choroidal vasculature and in the RPE, leading the formation of SRF [54].

A retrospective study showed that in 25 patients with CSC with *H. pylori* infection, a successful eradication of this bacterium led to a more rapid resolution of SRF in in comparison with 25 untreated patients who did not have an *H. pylori* infection [55]. Another study reported that *H. pylori* eradication in patients with aCSC did not improve BCVA and SRF reabsorption, but it increased the central retinal sensitivity [56].

In addition, another prospective, randomized clinical study on 33 patients with aCSC and *H. pylori* infection reported an improvement in BCVA and retinal sensitivity (measured by automated static perimetry) after *H. pylori* eradication [52].

To date, the evidence reporting *H. pylori* as a possible major risk for CSC is still limited and further clinical studies should be performed in order to assess *H. pylori* eradication benefits for treating CSC.

### 3.8. Ketoconazole

Ketoconazole is a synthetic imidazole exerting anti-fungal properties, but also displaying an anti-glucocorticoid action; for this last reason, it has been thought to be useful for treating CSC [57].

In a retrospective study, five patients with CSC were treated with oral ketoconazole 600 mg per day for 4 weeks. Although ketoconazole lowered endogenous cortisol levels, no significant differences were found in BCVA and OCT parameters [58].

In another non-randomized control study, ketoconazole at 200 mg/day was administered to 15 patients with CSC for 4 weeks. Additionally, in this study, no differences were reported in BCVA and OCT findings between the ketoconazole-treated patients and controls [59].

Thus, more studies are needed in order to consider ketoconazole as a viable option for treating CSC.

### 3.9. Methotrexate

Methotrexate is an antimetabolic and immunosuppressive agent commonly adopted for treating inflammatory diseases, especially in rheumatology (such as rheumatoid arthritis); considering its interaction with steroid receptors, this drug was postulated to be effective for CSC treatment [60].

In an interventional, prospective clinical trial 23 consecutive patients with cCSC were treated with oral methotrexate at 7.5 mg/week for 12 weeks. The average BCVA increased from 20/40 at baseline to 20/30 at the third month and 20/28 at the sixth month (*p* = 0.002 and 0.003, respectively). Furthermore, CMT decreased from 375 μm at baseline to 278 and 265 μm at the third and sixth months (*p* = 0.002 and 0.007, respectively) [61].

Another retrospective study examined nine patients with cCSC treated with low-dose methotrexate for an average of 89 days. It was found that 83% of them achieved complete resolution of SRF after an average treatment duration of 12 weeks [60].

A further control study should provide evidence about its clinical efficacy for treating CSC; nonetheless, methotrexate-related severe side effects, including bone marrow suppression and renal, hepatic and pulmonary toxicity represent an important limitation for considering this drug a first-line treatment option.

### 3.10. Rifampicin

Rifampicin is primarily adopted for the treatment of tuberculosis and other microbial infections; however, for its effect on cytochrome P450 3A4 induction and on the metabolism of endogenous steroids, rifampicin has been postulated to be effective also for treating CSC [62].

In a prospective pilot study, 12 patients with cCSC were treated with oral rifampicin at 300 mg twice per day for 3 months and were followed-up for 6 months. They found an increase in BCVA from 20/60 at baseline to an average of 20/50 at month 3 (*p* > 0.05). Moreover, CMT decreased by 25.3%, 21.2% and 21% by months 1, 2 and 3, respectively (*p* < 0.05); two patients stopped the treatment protocol due to rifampicin-related adverse events (cholelithiasis and systemic hypertension) [63].

In a case report, the complete resolution of SRF in a patient with cCSC was reported after 1 month of rifampicin treatment [64].

In a retrospective study on nine eyes of eight patients with cCSC treated with oral rifampicin, 44% of them showed complete resolution of SRF by FA after 10 months; by contrast, those patients with a diffuse leakage at FA did not improve [65].

In addition, Khan et al. reported in an observational study on 38 eyes of 31 patients with CSC an improvement in mean BCVA from 0.56 to 0.47 (*p* < 0.001) 4 weeks after 300 mg oral rifampicin treatment [66].

Importantly, rifampicin-associated side effects should not be neglected; in a case report, Nelson et al. documented a case of rifampicin-related hepatotoxicity in a patient with cCSC [67].

Given the variable results reported and the risk of severe side effects, rifampicin should not be considered a first-line treatment option for CSC treatment.

## 4. The Role of Anti-Vascular Endothelial Growth Factor (anti-VEGF) Agents

Several preclinical studies have suggested that anti-vascular endothelial growth factor (VEGF) agents are effective in reducing the permeability and proliferative activity of choroidal endothelial cells. In fact, these drugs are thought to upregulate the tight junctions between endothelial cells and the reduction of vascular fenestrations [68]. To date, no clear evidence about increased levels of VEGF in the aqueous humor of patients with CSC has been reported yet [69,70]. As a matter of fact, previous authors demonstrated that the aqueous humor level of VEGF in patients with chronic CSC was higher than that in patients with acute CSC without significance, and that there was a significant correlation between the duration of symptoms and the VEGF level in the aqueous humor [70].

There are four anti-VEGF agents approved by the USA FDA and EMA for treating retinal diseases and in particular wet age-related macular degeneration (w-AMD): pegaptanib sodium (Macugen^®^, Kirkland, QC, Canada), ranibizumab (Lucentis^®^ 2013 Informa UK. Genentech, South San Francisco, CA, USA/Roche, Basel, Switzerland), aflibercept (AFL, Eylea^®^, Regeneron^®^, Tarrytown, NY, USA) and brolucizumab (Beovu^®^, 2019 approved only by FDA) (42); in addition, bevacizumab is commonly used as an “off-label” intravitreal treatment option for retinal diseases [71,72]. Several reports confirmed that, and further revealed the contradictory efficacy of anti-VEGF in the treatment of CSC in the absence of CNV.

In a prospective study by Kim et al., 20 patients with cCSC were randomly assigned to ranibizumab intravitreal injections or placebo during a 6-month follow-up period. They found a significant increase in BCVA and a reduction in CMT in both the groups, with a faster complete resolution of neurosensory retinal detachment in the ranibizumab group than controls (13.0 ± 3.1 vs. 4.2 ± 0.9 weeks; *p* < 0.001) [73]. In another prospective control study, it was reported that 80% of 15 patients with CSC undergoing bevacizumab intravitreal injections showed complete resolution of SRF, as opposed to 53% of the 15 subjects in the control group [74]. A prospective, randomized study of 16 patients with cCSC treated either with low-fluence PDT or intravitreal injections of ranibizumab was recently performed. After a single session of PDT, six eyes (75%) in the low-fluence PDT group showed a complete resolution of SRF, as compared with only two (25%) eyes in the ranibizumab group. Moreover, a significant reduction of the foveal thickness was reported in the low-fluence PDT group (74.1 ± 56.0 to −35.4 ± 44.5 μm, *p* = 0.017), but not in the ranibizumab group (26.3 ± 50.6 μm to −23.1 ± 56.5 μm, *p* = 0.058) [75]. The same authors in another paper showed a complete resolution of SRF in only 13% of cCSC eyes in the ranibizumab group after 12 months, as opposed to 89% of them in the low-fluence PDT group [76]. In the CONTAIN study, patients with cCSC treated with aflibercept showed a significant resolution of SRF in 50% of the them, but no significant effect on BCVA, suggesting a possible better clinical efficacy for improving anatomical rather than functional outcomes [77].

Importantly, anti-VEGF drugs are expected to play a beneficial role in those patients with CSC complicated by secondary choroidal neovascularizations (CNVs) [12,69]. In this regard, a retrospective study on 46 eyes of 43 patients with cCSC complicated with CNVs and treated with anti-VEGFs, reported that 26% of them experienced an improvement greater than three lines, 41.3% of them has a stable vision (within ±1 line) and 16% of them showed a loss in BCVA greater than three lines. No adverse events were found and the average injections number was 4.45 ± 4.1 [78]. Recently it has been demonstrated with OCT-angiography, the presence of CNV growing under flat and undulated RPE detachment in patients with cCSC with SRF. However, it is not completely understood whether those type 1 CNVs are really responsible for the exudation, or whether they might be considered a compensatory effect to a subtle and chronic hypossia secondary to a sick choroid. Recently, researchers showed that in patients with cCSC, CNV demonstrated by OCT angiography responded well to bevacizumab injections [79].

However, although some promising results have been shown for treating patients with CNVs secondary to CSC, further larger-scale randomized clinical trials should provide more evidence in this direction.

## 5. Photodynamic Therapy (PDT)

Photodynamic therapy was introduced as an off-label treatment modality for CSC in 2003. This technique works by stimulating with a nonthermal infrared laser light (wavelength of 689 nm) verteporfin, a photosensitive dye injected intravenously, which accumulates in the altered choroidal vessels; then, verteporfin photo-stimulation leads to oxidative damage and remodeling of the choroidal microvasculature, causing ultimately SRF reabsorption [80]. Importantly, due to PDT high selectivity, the retinal photoreceptors layer is usually spared [81].

In the pivotal study, Yannuzzi et al. analyzed the effect of ICG-guided full-dose (6 mg/m^2^) PDT in 20 eyes of patients with cCSC. The authors found the complete resolution of macular detachments in 12 patients and incomplete resolution in another eight eyes. After a 6-week follow-up period, the mean BCVA increased significantly by 0.55 lines; no procedure-related adverse events were reported [82]. Similarly, Cardillo Piccolino et al. found that macular exudation resolved completely in 81% of the patients with cCSC after PDT treatment [83].

In a subsequent non-randomized, multicenter study, 82 eyes of 72 patients with cCSC were treated with full-dose PDT. They reported a complete resorption of SRF in 100% of the patients and an increase in BCVA (1.9 ± 2.4 Snellen lines) after an average follow-up period of 12 months. In the same study, reactivation of the disease was found only in two eyes (2%) [84].

In another multicenter, prospective, non-randomized study, 42 patients with cCSC were enrolled and treated with either ICGA-guided full-fluence PTD or low-fluence PDT. They found a complete reabsorption of SRF in 15 eyes in the full-fluence and 21 eyes in the low-fluence PDT groups (79% vs. 91%; *p* = 0.5). In addition, BCVA improved significantly in both the groups in all times (*p* < 0.01) [85].

Full-dose PDT has been reported not to be free from possible adverse events, including focal RPE losses, treatment-related CNVs, chronic hypoperfusion of the choroid and pigmentary changes [86].

In this regard, different PTD algorithms have been investigated, in order to decrease the probability of procedure-related adverse events. Verteporfin can be administered intravenously either at full-dose (6 mg/m^2^) or at half-dose (3 mg/m^2^); moreover, the applied light at the wavelength of 689 nm can be modulated either at a fluence of 50 J/cm^2^ or at half-fluence (25 J/cm^2^); lastly, treatment duration can be 83 s (full-time PTD) or 42 s (half-time PDT) [7].

In a retrospective study on 64 eyes from 60 patients with cCSC, 36 eyes were treated with low-fluence PDT (25 J/m^2^) and 28 eyes with half-dose verteporfin PDT (3 mg/m^2^). It was described that 91.6% of the eyes in the low-fluence group and 92.8% of them in the half-dose one had full reabsorption of SRF (*p* = 0.703). Moreover, BCVA improved, respectively, by 7.4 letters and 4.8 letters in the low-fluence and half-dose groups (*p* = 0.336) [87].

Differently, in a retrospective, multicenter study on 56 patients with cCSC, Nicolò et al. compared the clinical efficacy between 28 patients treated with half-dose PDT and 28 treated with half-fluence PDT. They reported a significant increase (*p* < 0.001) in average logMAR. BCVA improved significantly in the half-fluence group (from 0.187 (±0.187) to 0.083 (±0.164)) and in the half-dose group (from 0.126 (±0.091) to 0.068 (±0.091)) at 12 months, without a significant difference between the two treatment regimens. Moreover, after 1 month, complete reabsorption of SRF was found in 61.3% and 86.2% of the eyes in half-fluence and half-dose PDT groups, respectively (*p* = 0.04). After 12 months, full resolution of SRF was obtained in 83.9% and 100% of the eyes in half-fluence and half-dose PDT groups, respectively (*p* = 0.0529). Results from this study suggested that half-dose PTD may achieve more rapid and lasting resolution of the fluid as compared with half-fluence PDT [88].

Other studies reported that half-fluence PDT had a similar clinical efficacy to full-fluence PDT, and likewise, the half-dose PDT achieved comparable outcomes to the full-dose PDT [89,90].

Shiode et al. revealed that also halving the PDT irradiation time for treating patients with cCSC has led to similar clinical outcomes to adopting the half-dose PDT regimen [91]; these results were subsequently confirmed by Liu et al. in another study [92].

Furthermore, the multicenter PLACE trial revealed that a half-dose of PDT was superior to high-density subthreshold micropulse laser treatment, another laser technique commonly adopted for treating cCSC [93].

With regard to the optimal PDT dosage, it was found that half-dose PDT was more effective than a 30% dosage, with 95% of patients obtaining a complete SRF resolution, as opposed to 75% in the 30% dosage group [94].

Overall, PDT may be considered a safe procedure; in fact, Silva et al. reported in a 4-year follow-up safety study that full-dose PDT was not associated with the onset of side effects [95]; however, a meta-analysis revealed that patients treated with full-dose PTD had a greater percentage of side effects compared to the placebo group (22–42% vs. 16–23%), including abnormal vision, decreased vision and visual field defects [96]. Given this background, a reduced-setting PDT (half-dose, half-fluence and half-time PDT) has been largely employed in the clinical practice for the treatment of CSC, showing overall a good tolerability profile and the lack of any serious procedure-related adverse events [93].

However, it should be repeated that PDT might be ineffective or partially effective in inducing the resolution of the fluid in advanced forms of cCSC, associated with posterior cystoid retinal degeneration [97] (Table 1).

## 6. Conclusions

Central serous chorioretinopathy is a common cause of visual impairment in younger people, typically affecting patients aged 20 to 60 [98]. Although this disease is typically subdivided into an acute and a chronic form (aCSC and cCSC), there is still no unanimous consensus about its classification and treatment. While aCSC in many cases may resolve spontaneously and its treatment can be deferred, in some cases half-dose (or half-fluence) PDT has shown to be effective in accelerating SRF reabsorption. Importantly, large, multicenter, randomized clinical studies, including the PLACE trial, have suggested that half-dose PDT could represent a first-line treatment option for CSC, having been proven to be superior to high-density subthreshold micro pulse laser treatment [88,93].

Several oral medications have been investigated for the management of cCSC as possible non-invasive, first-line treatment options, including antioxidants, aspirin, methotrexate, ketoconazole, rifampicin, finasteride and others; however, to date, little evidence about their potential clinical efficacy has been provided, due in part to the few studies performed, their small sample sizes and their non-randomized nature [28]. Among them, mineralocorticoid receptor antagonists, and in particular eplerenone, were considered the most promising agents in this setting; however, the recent, multicenter, double-blind randomized VICI trial has demonstrated that eplererone is not superior to placebo for improving BCVA in patients with cCSC [26]. Thus, results from this study suggested that eplerenone should not considered as first-line, non-invasive treatment options for cCSC.

Lastly, anti-VEGF agents have been employed in clinical setting for treating CSC; to date, they have shown promising results only for the subgroup of patients with CNVs secondary to CSC [12].

In conclusion, further larger scale and well-structured clinical tries should provide more evidence about the clinical efficacy of oral medications for treating cCSC; to date, half-dose PDT is considered by most of the authors the gold-standard treatment for central serous chorioretinopathy.

## Figures and Tables

**Table 1 pharmaceuticals-13-00264-t001:** Levels of evidence of the main pharmacological treatment options investigated for the treatment of central serous chorioretinopathy.

Treatment Option	Level of Evidence	Classes of Recommendations	Clinical Notes
Photodynamic therapy (PTD)	A	I	First line treatment option for cCSC, safe procedure
Mineral corticoid and glucocorticoid-receptor oral antagonists	A	III	RCTs have shown no superiority to placebo
Antioxidants	B	III	Further evidence needs to be provided
Aspirin	C	IIb	Further evidence needs to be provided
Beta-blockers	C	III	Further evidence needs to be provided
Carbonic anhydrase inhibitors	C	III	Further evidence needs to be provided
Finasteride	C	III	Further evidence needs to be provided
*Helicobacter pylori* eradication	C	IIb	Further evidence needs to be provided
Methotrexate	C	III	Non-negligible side effects, further evidence needs to be provided
Melatonin	C	IIb	Further evidence needs to be provided
Finasteride	C	III	Further evidence needs to be provided
Ketconocazole	C	III	Further evidence needs to be provided
Anti-VEGF agents	A	I	Recommended for those patients with CNVs secondary to cCSC
Rifampicin	C	III	Severe drug-related side effects

cCSC = chronic central serous chorioretinoapthy; CNVs = choroidal neovascularizations; VEGF = vascular endothelial growth factor. **Levels of evidence:** A = strong evidence base: two or more high-quality studies; B = moderate evidence base: at least one high-quality study or multiple moderate-quality studies; C = limited evidence base: at least one study of moderate quality; I = insufficient evidence: evidence is insufficient or irreconcilable. **Classes of Recommendations:** Class I = evidence and/or general agreement that a given treatment or procedure is beneficial, useful, effective. Recommended; Class II = conflicting evidence and/or a divergence of opinion about the usefulness/efficacy of the given treatment or procedure; Class IIa = weight of evidence/opinion is in favor of usefulness/efficacy—should be considered; Class IIb = Usefulness/efficacy is less well established by evidence/opinion—may be considered; Class III = evidence or general agreement that the given treatment or procedure is not useful/effective, and in some cases may be harmful—not recommended.

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
