# Peer review of "Current Pharmacological Treatment Options for Central Serous Chorioretinopathy: A Review"

_pharmaceuticals, 2020, doi:10.3390/ph13100264_

Round 1
Reviewer 1 Report
The manuscript is well written; I did highly appreciate both the authors’ expertise and the flow of the text.
Abstract:
“….non-invasive treatment options SUCH AS XXX is attracting interest….”
PLACE; VICI, please spell out abbreviations when introducing them for the 1st time.
Introduction:
“….psychosocial stressors, a 35 type A personality, trait anxiety (Bazzazi et al., 2015), …”
“…..some randomized clinical trials have shown that PDT and subthreshold retinal laser treatment are the most promising treatment options currently available;”; here, the authors might describe a typical study; on doing so, the reader has an idea about how such studies could be performed also in the future.
“….given its more conservative approach”; what do you mean with ‘conservative approach’?
“….increased levels of cortisol and the dysregulation in the endogenous MR levels reported in patients with CSC (12).” Of note, such increased cortisol levels reflect an altered hypothalamic-pituitary-adrenocortical (HPA) axis activity as often observed in individuals with symptoms of depression, anxiety and stress (Holsboer and Ising, 2010; Jahangard et al., 2019; Miller et al., 2007)”. Given this background, could antidepressants be a further option?
As a general comment, try to report studies on humans and not on animals, as the reader might question the transferability of such study results.
The paragraphs on eplerenone and spironolactone were particularly well written. Congrats on you. Actually, this statement holds true also for the following paragraphs.
Table 1 provides a quick and concise overview of the current state-of-the-play.
References
Bazzazi, N., Ahmadpanah, M., Akbarzadeh, S., Seif Rabiei, M.A., Holsboer-Trachsler, E., Brand, S., 2015. In patients suffering from idiopathic central serous chorioretinopathy, anxiety scores are higher than in healthy controls, but do not vary according to sex or repeated central serous chorioretinopathy. Neuropsychiatric disease and treatment 11, 1131-1136.
Holsboer, F., Ising, M., 2010. Stress hormone regulation: biological role and translation into therapy. Annual review of psychology 61, 81-109.
Jahangard, L., Mikoteit, T., Bahiraei, S., Zamanibonab, M., Haghighi, M., Sadeghi Bahmani, D., Brand, S., 2019. Prenatal and Postnatal Hair Steroid Levels Predict Post-Partum Depression 12 Weeks after Delivery. Journal of clinical medicine 8(9).
Miller, G.E., Chen, E., Zhou, E.S., 2007. If it goes up, must it come down? Chronic stress and the hypothalamic-pituitary-adrenocortical axis in humans. Psychol Bull 133(1), 25-45.
Author Response
The manuscript is well written; I did highly appreciate both the authors’ expertise and the flow of the text.
Abstract:
“....non-invasive treatment options SUCH AS XXX is attracting interest....” PLACE; VICI, please spell out abbreviations when introducing them for the 1st time.
We spelled out the abbreviations in the abstract as suggested by the reviewer
Introduction:
“....psychosocial stressors, a 35 type A personality, trait anxiety (Bazzazi et al., 2015), ...”
We added thee reference suggested by the reviewer and modified the sentence as suggested
“.....some randomized clinical trials have shown that PDT and subthreshold retinal laser treatment are the most promising treatment options currently available;”; here, the authors might describe a typical study; on doing so, the reader has an idea about how such studies could be performed also in the future.
We thank the reviewer for this comment. We better described a typical study revealing the clinical efficacy of photodynamic therapy for treating CSC as suggested.
“....given its more conservative approach”; what do you mean with ‘conservative approach’?
The term ‘conservative’ deals with all the non-invasive and non- surgical procedures. We better explained it in the text.
1
“....increased levels of cortisol and the dysregulation in the endogenous MR levels reported in patients with CSC (12).” Of note, such increased cortisol levels reflect an altered hypothalamic-pituitary-adrenocortical (HPA) axis activity as often observed in individuals with symptoms of depression, anxiety and stress (Holsboer and Ising, 2010; Jahangard et al., 2019; Miller et al., 2007)”. Given this background, could antidepressants be a further option?
We thank the reviewer for this interesting insight. We modified the manuscript discussing the possible role of antidepressants drugs for treating CSC; however, to date, no antidepressants drugs have been investigated for treating CSC.
As a general comment, try to report studies on humans and not on animals, as the reader might question the transferability of such study results.
We thank the reviwer for this comment; however, we deem that in order to better understand the pathophydiology of CSC also studies on animals may be helpful and explanatory and therefore necessary to transfer results from preclinical studies to human models.
The paragraphs on eplerenone and spironolactone were particularly well written. Congrats on you. Actually, this statement holds true also for the following paragraphs.
We thank the reviewer for the comment
Table 1 provides a quick and concise overview of the current state-of-the-play.
We thank the reviewer for the comment.
References
Bazzazi, N., Ahmadpanah, M., Akbarzadeh, S., Seif Rabiei, M.A., Holsboer-Trachsler, E., Brand, S., 2015. In patients suffering from idiopathic central serous chorioretinopathy, anxiety scores are higher than in healthy controls, but do not vary according to sex or repeated central serous chorioretinopathy. Neuropsychiatric disease and treatment 11, 1131- 1136.
Holsboer, F., Ising, M., 2010. Stress hormone regulation: biological role and translation into therapy. Annual review of psychology 61, 81-109.
Jahangard, L., Mikoteit, T., Bahiraei, S., Zamanibonab, M., Haghighi, M., Sadeghi Bahmani, D., Brand, S., 2019. Prenatal and Postnatal Hair Steroid Levels Predict Post- Partum Depression 12 Weeks after Delivery. Journal of clinical medicine 8(9).
Miller, G.E., Chen, E., Zhou, E.S., 2007. If it goes up, must it come down? Chronic stress and the hypothalamic-pituitary-adrenocortical axis in humans. Psychol Bull 133(1), 25-45.
Reviewer 2 Report
The paper by Nicolò, M. is interesting and also ideal for the journal 'Pharmaceuticals'. I have a couple of comments to improve the article:
(1) If the whole article is about central serous chorioretinopathy, why don't the authors expand more on the introductory section of this disease? Also, why the authors mention the imaging modalities in the 4th paragraph of the introduction? Imaging modalities are an important part of the diagnostics but the authors should explain it later in the manuscript.
(2) Table 1 is interesting and informative for any researcher working in the field. But, what I still do lack in the article is any therapeutic challenge that is faced by the field of clinicians/ researchers?
Author Response
The paper by Nicolò, M. is interesting and also ideal for the journal 'Pharmaceuticals'. I have a couple of comments to improve the article:
(1) If the whole article is about central serous chorioretinopathy, why don't the authors expand more on the introductory section of this disease? Also, why the authors mention the imaging modalities in the 4th paragraph of the introduction? Imaging modalities are an important part of the diagnostics but the authors should explain it later in the manuscript.
We thank the reviewer for this comment. We think that the main pathophysiological and epidemioloigcal aspects of CSC were explained in the introduction. We highlighted the important role of diagnostic modalities for the management of CSC as suggested by the reviewer.
(2) Table 1 is interesting and informative for any researcher working in the field. But, what I still do lack in the article is any therapeutic challenge that is faced by the field of clinicians/ researchers?
We thank the reviewer for this comment; as discussed in the conclusion paragraph, there are still many therapeutic challanges about the role of oral medications for treating CSC, given the lack of large, randomized, multicenter studies. In the near future, larger and well-strucutured studies should provide more evidence in this regard.